# An Interesting Molecule: γ-Aminobutyric Acid. What Can We Learn from Hydra Polyps?

**DOI:** 10.3390/brainsci11040437

**Published:** 2021-03-29

**Authors:** Paola Pierobon

**Affiliations:** Institute of Applied Sciences and Intelligent Systems E. Caianiello, CNR, Via Campi Flegrei 34, 80078 Pozzuoli, Italy; paola.pierobon@isasi.cnr.it

**Keywords:** gamma-aminobutyric acid, GABA shunt, gut–brain axis, horizontal gene transfer, *Hydra*

## Abstract

Neuronal excitability is controlled primarily by γ-aminobutyric acid (GABA) in the central and peripheral nervous systems of vertebrate as well as invertebrate organisms. Besides its recognized neurotransmitter functions, GABA also plays a fundamental role in neurogenesis and synaptogenesis during embryonic development. In addition, GABAergic mechanisms are also involved in disorders of various peripheral tissues, ranging from diabetes to hypothyroidism to inflammatory responses. The discovery of the molecule and the history of its biosynthetic pathways in vertebrate and invertebrate phyla are summarized here. The occurrence and distribution of GABA, GABA-synthesizing enzymes, and receptors to GABA in the freshwater polyp *Hydra vulgaris* (Cnidaria: Hydrozoa), endowed with an early evolved nervous system, are discussed in relation to possible interactions with the microbiota, a stable component of Hydra polyps; their contribution to the evolution of nervous systems through microbe–neuronal interactions is proposed.

## 1. Introduction

Invertebrate research has greatly contributed to our present understanding of brain function. In the second half of the last century, comparative studies carried out in various invertebrate organisms added or, in several cases, provided evidence of new biological and physiological properties of neurons and, more importantly, indicated that the elementary unit of nervous systems, the nerve cell, shares similar or equal structural and functional mechanisms in vertebrates and invertebrates, even in distantly related species [1]. In more recent years, molecular biology studies, the sequencing of human genome, and the following explosion of genome sequencing of several species prompted comparative genomic and phylogenetic analyses of organisms and of their cellular and molecular components, including the nervous systems [2]. The wealth of new available information on organs and structures is presently stimulating different theoretical approaches, roughly development and/or behavior versus molecular phylogenetics, deep homology versus convergent evolution, towards a clearer understanding of generative processes and specification mechanisms in living organisms [3].

Here, I will attempt a brief outline of the contribution to these new exciting developments coming from new and old knowledge of the biological functions of γ-aminobutyric acid (GABA), the major inhibitory neurotransmitter in the vertebrate central nervous system (CNS) and from studies on GABAergic systems in the freshwater polyp *Hydra vulgaris* Pallas (Cnidaria: Hydrozoa).

## 2. The Discovery of GABA and Its Biological Functions

GABA is a four-carbon amino acid found in plants as well as prokaryotic and eukaryotic organisms as a significant component of the cellular free amino acid pool. GABA is an atypical amino acid, in that it is not used for protein synthesis, since the amino group is attached to the gamma carbon, differently than in protein-forming alpha amino acids. Synthesized in 1883, it was known for many years only as a plant and microbe metabolic product. Only in 1950, it was first identified as a normal constituent of mammalian CNS [4]. Around ten years later, it was shown that GABA acted as an inhibitor on crayfish and crab muscle fibers [5,6]. In the 1960s, it was finally established as the principal inhibitory transmitter in the invertebrate and vertebrate nervous systems [7,8], acting through specific membrane receptors, either ionotropic or ligand-gated ion channels (LGIC), the GABA_A_ receptors [9], or metabotropic, G-protein-coupled receptors (GPCR), named GABA_B_ receptors [10].

GABA acts on neuronal excitability by regulating chloride ions’ entry into the cell; the direction of Cl^−^ flow depends on extracellular Cl^−^ levels. In the adult brain, GABA is mainly inhibitory, although, in axo-axonic cells in cortical microcircuits, it can also be excitatory [11]. It has been proposed that during brain development, GABA’s role changes from excitatory [12] to inhibitory due to a developmental switch in the molecular machinery controlling intracellular Cl^−^ concentration [13]. More recent studies, however, have cast doubt on the excitatory GABA theory, suggesting that experimental artifacts and/or neuronal damage may determine the discrepancies between results obtained by in vivo studies or in vitro slice preparations and pointing to the need of non-invasive new techniques to address these issues [14]. 

Besides its function in cell signaling, GABA also has neurotrophic functions [15], regulating the proliferation, migration, and differentiation of neural stem cells by neurotrophic factor expression [16], neurite elongation, and synapse formation [17]. In addition, GABA signaling through GABA_A_ receptor activation may play a role in the generation of embryonic and peripheral neural crest stem cells by hyperpolarization and decrease in proliferation [18,19].

Altered GABAergic functions are responsible for neurological and psychiatric disorders related to hyperexcitability, including epilepsy, drug dependence, anxiety, bipolar disorders, as well as motor coordination diseases such as tardive dyskinesia and Huntington’s and Parkinson’s diseases [20]. GABA receptors, and especially GABA_A_Rs, can be activated or modulated by a plethora of allosteric ligands, among them benzodiazepines and derivatives, popular anxiolytics. Development of new drugs for an improved treatment of these illnesses is a very active field of research.

Parallel to studies of the role of GABA and GABAergic system in neuronal signaling, research efforts focused on the occurrence and functions of GABA outside the brain showed that GABA is found in several peripheral tissues and organs, where it participates in the regulation of various cellular processes, from replication of the insulin-producing β cells of the pancreas, to regulation of cytokine secretion, to stimulation of regulatory immune responses through T-lymphocytes; here, I will focus on the effects of GABA on the enteric nervous system [21].

Interactions between the brain and the digestive system in health and disease are well known, leading to the development of the concept of a gut–brain axis, or the biochemical signaling between the CNS, the neuroendocrine and neuroimmune systems, and the intestine; it has received renewed interest since the discovery of the enteric nervous system [22]. Microbiota, the bacterial complement in our bodies, is emerging as a key regulator of the gut–brain axis and its interactions with the immune system, suggesting the establishment of a microbiota–gut–brain axis as an extension of the gut–brain axis concept [23]. Host–microbe interactions affect many levels of complexity, from intercellular communication to systemic signaling in various organs and systems, including CNS. Changes in the composition of the microbiota may affect metabolic processes such as immunomodulation as well as behavior and cognition.

GABA has been implied in many diseases associated with malfunctioning of the gut–brain axis, ranging from stress-related disorders such as irritable bowel syndrome to neurodevelopmental disorders such as autism. Exogenous GABA as a nutritional supplement could affect the enteric nervous system, which in turn would stimulate endogenous GABA production across the blood–brain barrier [24,25].

## 3. GABA Biosynthesis and Metabolic Pathways

The metabolic pathway of GABA biosynthesis was described in detail in the early 1950s. In the following years, the discovery of the neurotransmitter role of GABA and the identification of the related receptors directed research efforts towards the molecular characterization of receptor subunits, the physiological and pharmacological properties of different subunit assemblies, and the kinetics of ligand binding, GABA production slowly sliding out of view. Nonetheless, readers may find useful a short reminder of this basic metabolic process. 

GABA is formed in vivo by transamination of α-ketoglutarate, derived from glucose metabolism in the Krebs cycle, into L-glutamic acid by the enzyme α−ketoglutarate transaminase (GABA-T). A second enzyme, glutamic acid decarboxylase (GAD), catalyzes the decarboxylation of glutamic acid into GABA. Conversely, in the presence of α-ketoglutarate, GABA is degraded by GABA-T into succinic semialdehyde and glutamic acid; succinic semialdehyde can be oxidized into succinic acid, which re-enters the Krebs cycle, while glutamic acid restores the supply for new GABA synthesis [26,27]. The process by which the GABA reserve is produced and maintained is called the GABA shunt [28].

In mammalian neurons, this metabolic pathway appears to be the primary source of the neurotransmitter, since, in general, GABA does not cross the blood–brain barrier [29] (but see [25]). Accordingly, GAD, the key enzyme involved in GABA synthesis, is selectively expressed in cells that use GABA as a neurotransmitter [30]; however, expression of GAD and some GABA receptor subunits has been demonstrated in non-neural tissues [31]. In mammals, GAD exists in two isoforms of different molecular weight, GAD67 and GAD65, encoded by two different genes, *GAD1* and *GAD2*, respectively, which are differentially regulated, though nearly every GABA-producing cell contains both forms of GAD. GAD67 predominates early in development; in the mature neuron, GAD67 is present in both terminals and the cell body, where it may subserve a non-synaptic, intracellular GABA pool. In contrast, GAD65 is usually expressed later in development and is primarily localized to nerve terminals [32]. The newly produced GABA is transported and packaged into synaptic vesicles with the help of GABA transporter proteins [33].

Besides plants and bacteria, GABA and the GABA shunt enzymes are found in many invertebrate species, namely worms, mollusks, crustaceans, insects and other arthropods, echinoderms, as well as cephalochordates and tunicates. To my knowledge, a systematic review of the occurrence of GAD in invertebrates has not been attempted, and it is beyond the purpose of this paper. A phylogenetic tree drawn on the basis of sequence homology of GAD subunits of different origin suggests convergent evolution of the enzyme [34]. Some authors suggest that the evolution of GABA_A_ receptors must have depended on the evolution of GAD [35].

GABA and GAD are also found in non-bilaterian phyla such as Porifera and Cnidaria. Sponges lack conventional nerves and muscles but respond to mechanical and chemical stimuli, possibly by paracrine signaling mechanisms; coordinated canal contraction behavior is regulated by pools of glutamate, glutamine, and GABA [36,37]. Cnidarian species display a variety of nervous systems of different anatomical and cellular complexity; GABA, GABA receptors, and, in some cases, GABA-related enzymes have been found in jellyfish, sea anemones, and gorgonian corals, where they are involved in intercellular signaling as well as neurogenesis, gametogenesis, and metamorphosis [38,39,40,41,42]. A synthetic description of current knowledge on the GABAergic system in prokaryotes and non-bilaterian eukaryotes is shown in Table 1.

As shown in Table 1, GAD isoforms and related genes, GABA-T, and different GABA transporters are found in several bacterial species [43] as well as in sponges [45]. The GLIC/ELIC membrane receptors recognize GABA as well as other transmitter molecules [44]. In sea anemone sensory neurons, co-localization of GABA and GAD immunoreactivity is observed [39]. Additional references.

## 4. GABA and *Hydra*: Available Evidence

In *Hydra* polyps, the neurons are connected to one another to form a net spreading homogeneously throughout the body, except at the head and foot regions, where the fibers are condensed into a circular nerve ring. The nerve net is continuously remodeled in the adult polyp by cell differentiation and migration along an oral–aboral axis. Current knowledge indicates that in *Hydra* species, the nerve net, one of the most primitive nervous systems to have evolved, shows a greater structural and functional complexity than previously acknowledged, modulating different behavioral responses through a variety of cellular effectors (reviewed in [46]). Neuronal signaling relies largely on neuropeptides, but increasing evidence of the occurrence of classical neurotransmitters is available [47]. Our group identified, characterized, and localized in *Hydra* tissues the receptors to inhibitory and excitatory amino acid neurotransmitters, GABA, glycine, and NMDA, that compare with mammalian ionotropic receptors by their biochemical and pharmacological properties [46]. Administration of exogenous ligands showed that these receptors appear to regulate pacemaker activities and their physiological correlates; in the live animal, they also affect the feeding behavior, namely duration and termination of the response elicited by reduced glutathione, with opposite actions of GABA and glycine or NMDA, respectively.

Receptors to GABA were identified in membrane preparations from *Hydra vulgaris*. The binding of [^3^H]GABA was specific (70% specific binding), reversible, and saturable. A Scatchard analysis of saturation data indicated the presence of only one population of binding sites with high affinity (K_D_ = 76 nM) and low capacity (B_max_ = 4.75 pmol/mg of protein). A systematic screening of GABA_A_ receptor allosteric modulators provided evidence that: (a) the benzodiazepine (BZ) ligands diazepam, clonazepam, and abecarnil enhanced [^3^H]GABA binding to *Hydra* membranes by 20–24%, effects abolished by the specific central BZ antagonist flumazenil. On the contrary, the peripheral BZ receptor ligand 4’chlorodiazepam failed to affect [^3^H]GABA binding to *Hydra* membranes; (b) the neuroactive steroids allopregnanolone and tetrahydrodeoxycorticosterone increased [^3^H]GABA binding to *Hydra* membranes with nanomolar potency and high efficacy, whereas the 3β-hydroxy epimer of allopregnanolone was ineffective; (c) the general anesthetics alphaxalone, barbiturates, and propofol similarly increased [^3^H]GABA binding.

Finally, competition experiments showed that [^3^H]GABA binding was completely inhibited by the GABA agonist muscimol and by the GABA_A_ receptor antagonist gabazine, whereas bicuculline was ineffective; baclofen, a GABA_B_ receptor antagonist, weakly (>30%) inhibited [^3^H]GABA binding [48].

The modulation of *Hydra* GABA receptor proteins by these various drugs *in vitro* correlated with their effects in living animals. GABA had an inhibitory role on muscle contraction: GABA, muscimol, and diazepam were shown to decrease the frequency of contraction burst pulses in the ectodermal pacemaker system, (GABA), and the frequency of rhythmic potentials in the endodermal rhythmic potentials system (GABA, muscimol, diazepam); none of the ligands affected tentacle pulse frequency, i.e., the electrical activity of the tentacles. Interestingly, the inhibitory action of GABA was suppressed by bicuculline, per se ineffective [49].

In the living polyp, GABA affects the duration of the feeding behavior, possibly acting on the muscle fibers that control mouth opening upon prey capture or exposure to reduced glutathione (GSH). Administration of 100 μM GABA significantly increased the duration of mouth opening induced by GSH. As predicted from biochemical experiments, GABA_A_R agonists and positive allosteric modulators mimicked or potentiated these effects of GABA in a dose-dependent manner. The classical GABA_A_ receptor antagonist bicuculline, per se inactive, was able to suppress the GABA-induced increase in response duration. Another GABA_A_ receptor antagonist, gabazine, suppressed the effects of GABA and GABA agonists in a 1–10 μM concentration range; in the absence of GABA, it shortened response duration. The specific Cl^−^ channel blockers, picrotoxin (10 μM) and t-butylbyciclophosphorothionate (TBPS) (1 µM), which per se shortened the duration of the response, also abolished the increase in response duration induced by GABA and counteracted the effects of muscimol, neurosteroids, and general anesthetics [48].

Thus, the apparent insensitivity to bicuculline shown in competition experiments was not supported by *in vivo* experiments. Similarly, GABA and the metabotropic GABA_B_R agonist baclofen significantly increased the rate of distant desmoneme (stinging cells) discharge in *Hydra* tentacles [50] by acting through GABA_B_ receptors [51]. In whole animals, baclofen in a 0.05–100 μM concentration range did not significantly modify the duration of the GSH-induced response; however, baclofen was able to reduce response duration in amputated heads, at 10-μM and 100-μM doses. The decrease was suppressed by the GABA_B_R antagonist phaclofen, per se ineffective at 10-μM or 100-μM concentrations both in isolated heads and in whole polyps [52].

Taken together, these data provide evidence that, besides putative metabotropic GABA receptors, a class of GABA receptor proteins, the ligand-gated ion channel (LGIC) superfamily is present in *Hydra* tissues whose pharmacological properties are typical of canonical GABA_A_Rs. An immunohistochemical study indicates that receptor subpopulations of different subunit composition, α3, β1, γ3, and δ subunits, are found in different body regions of *Hydra* polyps [53]. The occurrence of putative ionotropic GABA receptors with a high degree of structural complexity in a species endowed with an early evolved nervous system is an important contribution to phylogenetic studies on the origin of LGICs. The finding that the response to GSH is shortened by gabazine and by Cl^−^ channel blockers suggests a role of endogenous GABA in modulation of the response in physiological conditions.

## 5. The Microbiome of *Hydra*: Host–Microbe Interactions 

At the time that the prevalent research interests were directed towards understanding the cellular and molecular basis of growth, differentiation, and regeneration processes in *Hydra*, a group studying epithelial and stem cell lineages, led by Thomas Bosch, was becoming very concerned with bacterial “contamination”: were bacteria occasional intruders or stable symbionts? What was the bacterial contribution to cell metabolism? Did bacteria contribute to host physiology and behavior? How did polyps defend themselves from eventual bacterial aggression? The answer to these and other questions led to an impressive series of results showing that “the presence and structure of Hydra’s microbiota are critical for tissue homeostasis and health of the polyps…epithelia and components of the innate immune system play an active role in selecting the inhabitant microbiota via a complex genetic network” (reviewed in [54]). The recent discovery of a direct interaction of *Hydra* pacemaker neurons with the commensal microbiota, using components of the innate immune system, indicates the antiquity of the establishment of the microbiota–gut–brain axis [55]. Especially relevant to the presence of GABA is the finding that spontaneous body contractions of *Hydra*, essential to animal survival, are regulated by the microbiome, possibly through the secretion of small, water-soluble molecules [56]. In fact, it is interesting to note that at least one component of the *Hydra* microbiome, *Pseudomonas*, is a GABA-producing bacterial species [57]. 

## 6. GABA and *Hydra*: Open Questions

The data so far obtained suggest that GABAergic signaling regulates muscle fibers and the nerve net in *Hydra* tissues. However, several questions should be answered in order to establish a neurotransmitter role for GABA, such as the presence and cellular localization of GABA and its metabolic machinery, and related genes. The biochemical and functional characterization of GABA receptor proteins should enable the identification of the corresponding genes. Finally, the synaptic and/or eventual extrasynaptic localization of GABA receptors would help in understanding the regulatory role exerted by GABA on the electrical activity of the neuronal pacemakers and muscle fibers.

The occurrence of GABA itself in the nerve net and/or in tissues of *Hydra* polyps has not been established by our studies, apart from some preliminary experiments by gas chromatography/mass spectrometry analysis of tissue homogenates revealing the presence of three fragmentation peaks perfectly coincident with those obtained from synthetic GABA at the same time of elution; this result was reproduced with two different derivatization methods. However, at the time, we considered these findings inconclusive, since we were not able to confirm these findings by convincing autoradiographic localization of GABA in nerve cells. In fact, *Hydra* neurons are not clustered in ganglia or in an anatomically distinguishable brain but are dispersed in the two epithelial layers that comprise the polyp body; as a consequence, biochemical experiments need to be carried out on whole body homogenates. A stable bacterial community colonizes the body surface [54]; also, the epithelial cells contain a variety of bacteria in large intracellular vacuoles, as we discovered years ago when working on a protocol to prepare cell cultures for electrophysiological experiments. As a consequence, measuring GABA levels would require preparing germ-free polyps and/or screening the bacterial populations present in the tissues in order to evaluate their contribution to the extant GABA pool. Both issues were quite out of focus for us, since we were studying neurotransmission and related molecules, so, at the time, we chose a different approach, centered on transmitter biological actions and effectors. 

The molecular identification of GABA receptor proteins and genes in the *Hydra* genome is still in its infancy. A systematic analysis of the *Nematostella* genome [58] showed (a) the presence of several GABA_B_ receptors comparable to their vertebrate and invertebrate counterparts; (b) more GABA_A_ receptor sequences with low identity in residues important for GABA binding; (c) two apparent splice variants of GAD, with more conserved residues with prokaryote than with eukaryote GAD. A large number of inhibitory amino acid transporters were also identified; their sequence is highly conserved, with high residue similarity and identity with the GABA transporter GAT1 orthologues.

While these findings should be corroborated by genomic analysis of other cnidarian representatives—and, in particular, of *Hydra* species—they suggest a provocative hypothesis: if the initial GABA supply was provided to *Hydra* by its bacterial community, as inferred from the above-mentioned studies and hinted by the relative abundance of GABA transporter sequences, could the GAD gene(s) have entered the genome by horizontal transfer and later be stabilized as a functional component of the GABAergic machinery? I am not entirely alone in this thought: a recent study reports that transposable elements contribute to genome expansion in a subgroup of the genus, the brown hydras, to which *Hydra vulgaris* belongs, finding no evidence of genome duplication [59].

Transfer of genetic material between non-mating species is increasingly recognized as an important contribution to the evolution of eukaryotic genomes. Internalization of multicellular tissues, including the neural plate in vertebrates, is a crucial morphogenetic process in development and organ formation; the contribution of microbes to internalization of nervous systems has been proposed as a working hypothesis [60]. The role of bi-directional microbiota–neuron interactions at the molecular level opens new directions of research for our understanding of the many-faceted actions of GABA in animal organisms.

## Figures and Tables

**Table 1 brainsci-11-00437-t001:** Occurrence of GABA and GABA machinery in prokaryotes and non-bilaterian phyla.

	Bacteria	Sponges	Cnidaria
GABA metabolism	GAD, GAD genesGABA-T	GAD, GAD genesGABA-T	GADGABA-T
GABA sensors	GABA transportersGLIC, ELIC receptors	GABA transportersGABA_B_ receptors	VGATGABA receptors
plcGABA production and cellular localization	GABA	GABAchoanocytes, pinacocytes	GABASensory neuronsNerve fibers
Physiological role(s)	The GAD system contributes tosurvival in acidic environments by increase in internal pH and alkalinization of external fluids	GABAergic inhibitory regulation of water flow, body contraction, and feeding in response to external signals	Neuronal signalingPhotic perceptionRegulation of nematocyst dischargeModulation of neurogenesis, development
References		[36,37]	[38,39,40,41,42]
Additional references	[43,44]	[45]

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
