# Peer review of "An Interesting Molecule: γ-Aminobutyric Acid. What Can We Learn from Hydra Polyps?"

_brainsci, 2021, doi:10.3390/brainsci11040437_

Round 1

Reviewer 1 Report

This review summarized role of ganma-aminobutyric acid (GABA) molecule and its biosynthetic pathways in vertbrate and invertbrate phyla. The manuscript is well organized, very useful for a broad reader to quickly grasp the knowledge of GABA history.

However, the authors should reduce the number of references before acceptance.

Author Response

References have been reduced and formatted according to the journal style

Reviewer 2 Report

In this article the author first reviewed the GABA system of vertebrate - from GABA synthesis and metabolism to receptors and functions, followed by some recent findings in the polyp Hydra and other invertebrate species  suggesting possible presence of the GABA system. Then, the author proposed a new hypothesis that GABA-related microbe-neuronal interactions might be present in vertebrates (mammal).

However, by carefully reading this article, this reviewer still cannot understand what the new findings are in polyps and what is the author's hypothesis. As stated by the author, there is no convincing evidence indicating that GABA is present and functional in Hydra neurons. It is unclear how the findings of GABA receptor and GAD genes in Nematostella genome imply their presence in Hydra and what is the basis of the proposed microbe-neuronal interaction hypothesis?  Are the GABA system-related genes and proteins present in any bacteria or microbe? 

Again, this reviewer cannot find convincing evidence and rationale supporting the author's hypothesis. The author may need to substantially revise the manuscript to clearly tell readers what the new findings are in Hydra, what is the author's hypothesis, how it is related to GABA system, and what the author has learnt from Hydra polyps.                      

Author Response

a detailed overview of the presence of GABA receptors in Hydra cells and tissues has been added to the paper (lines 292 to 352). The issue of GABA occurrence in nerve cells has been further discussed (lines 353 to 362). The production of GABA by the bacterium Pseudomonas has been specified (lines 389-390)

Reviewer 3 Report

In the present opinion article Dr Pierobon provides and overview of the GABAergic system and its function in Hydra polyps. The presence of Hydra’s microbiota and its critical role for tissue homeostasis and health of the polyps are also discussed. Also proposes the hypothesis that initial GABA supply was provided to Hydra by its bacterial community, but GAD gene(s) have entered the genome by horizontal transfer.

This is an interesting topic that will be of interest to investigators working in the field of neuroscience. The manuscript is logically structured, and the previous literature that supports the hypothesis are well described.

Minor issues: There are some formatting problem and the referencing style is not supported by the journal.

Author Response

References have been reduced and formatted according to the journal style. The text has been checked for typing errors.

Round 2

Reviewer 2 Report

I can see some improvement in the revised manuscript, but my questions and concerns are still there, not addressed. This may be related to a mistake that the editorial office did not give the author my original comments in detail to author. Alternatively, they gave the author my comment to the editors.

The questions are: what is the evidence supporting the presence of GABA or GABA related genes or molecules in microbiota? What is the evidence supporting GABA interactions between microbe and hydra neurons?  The author provided evidence supporting the presence of inhibitory GABA signaling in hydra neurons. This is not surprising or a new finding since GABA signal molecules are often found in neurons in both vertebrates and invertebrates. This reviewer cannot find the evidence from this article supporting the proposed microbe-neuron hypothesis.

Author Response

1) Evidence of the presence in bacteria of GABA, its synthesizing enzymes, and related genes would be suited for an independent review or, better, a chapter in a microbiology book. Certainly, it cannot be covered in this short paper, nor can the relevant supplementary references be added to a list that I already had to shorten as requested by all reviewers. However, a sentence added in the previously revised version (lines 388-389) specifies that at least one bacterial specie present in Hydra microbiome produces GABA: the related reference has been added to the text 

2) GABA interactions between microbiome and neurons in Hydra are currently an expanding research field; the available results have been cited (ref 52)